# Proxy Convexity: A Unified Framework for the Analysis of Neural Networks Trained by Gradient Descent

**Spencer Frei**
Simons Institute for the Theory of Computing
University of California, Berkeley
frei@berkeley.edu

**Quanquan Gu**
Department of Computer Science
University of California, Los Angeles
qgu@cs.ucla.edu

## Abstract

Although the optimization objectives for learning neural networks are highly non-convex, gradient-based methods have been wildly successful at learning neural networks in practice. This juxtaposition has led to a number of recent studies on provable guarantees for neural networks trained by gradient descent. Unfortunately, the techniques in these works are often highly specific to the particular setup in each problem, making it difficult to generalize across different settings. To address this drawback in the literature, we propose a unified non-convex optimization framework for the analysis of neural network training. We introduce the notions of proxy convexity and proxy Polyak-Lojasiewicz (PL) inequalities, which are satisfied if the original objective function induces a proxy objective function that is implicitly minimized when using gradient methods. We show that stochastic gradient descent (SGD) on objectives satisfying proxy convexity or the proxy PL inequality leads to efficient guarantees for proxy objective functions. We further show that many existing guarantees for neural networks trained by gradient descent can be unified through proxy convexity and proxy PL inequalities.

## 1 Introduction

Understanding the ability of gradient-based stochastic optimization algorithms to find good minima of non-convex objective functions has become an especially important problem due to the success of stochastic gradient descent (SGD) in learning deep neural networks. Although there exist non-convex objective functions and domains for which SGD will necessarily lead to sub-optimal local minima, it appears that for many problems of interest in deep learning, across domains as varied as natural language and images, these worst-case situations do not arise. Indeed, a number of recent works have developed provable guarantees for GD and SGD when used for objective functions defined in terms of neural networks over certain distributions, despite the non-convexity of the underlying optimization problem [5, 3, 6, 22, 14, 16]. To date, however, there has not been a framework which could unify the variegated approaches for guarantees in these settings.

In this work, we introduce the notion of *proxy convexity* and demonstrate that many existing provable guarantees for learning with neural networks trained by gradient-based optimization fall into a problem satisfying proxy convexity. First, let us define the *learning problem* over a distribution $\mathcal{D}$, where the goal is to minimize the expected loss

$$\min_{w \in \mathcal{W}} F(w) := \mathbb{E}_{z \sim \mathcal{D}} f(w; z), \tag{1.1}$$

35th Conference on Neural Information Processing Systems (NeurIPS 2021).

where $\mathcal{W}$ is a parameter domain and $f(\cdot\,;z)$ is a loss function. We are interested in guarantees using the online SGD algorithm, which uses a set of i.i.d. samples $z_t \overset{\text{i.i.d.}}{\sim} \mathcal{D}$ and has updates given by

$$w_{t+1} = w_t - \eta \nabla f(w_t; z_t),$$

where $\eta > 0$ is a fixed learning rate. We now introduce the first notion of proxy convexity we will consider in the paper.

**Definition 1.1** (Proxy convexity). *We say that a function $f : \mathbb{R}^p \to \mathbb{R}$ satisfies $(g,h)$-proxy convexity if there exist functions $g, h : \mathbb{R}^p \to \mathbb{R}$ such that for all $w, v \in \mathbb{R}^p$,*

$$\langle \nabla f(w), w - v \rangle \geq g(w) - h(v).$$

Clearly, every convex function $f$ satisfies $(f, f)$-proxy convexity. We next introduce the analogy of proxy convexity for the Polyak–Łojasiewicz (PL) inequality [23].

**Definition 1.2** ($g$-proxy, $\xi$-optimal PL inequality). *We say that a function $f : \mathbb{R}^p \to \mathbb{R}$ satisfies a $g$-proxy, $\xi$-optimal Polyak–Łojasiewicz inequality with parameters $\alpha \in (0, 2]$ and $\mu > 0$ (in short, $f$ satisfies the $(g, \xi, \alpha, \mu)$-PL inequality) if there exists a function $g : \mathbb{R}^p \to \mathbb{R}$ and scalars $\xi \in \mathbb{R}$, $\mu > 0$ such that for all $w \in \mathbb{R}^p$,*

$$\|\nabla f(w)\|^{\alpha} \geq \frac{1}{2}\mu\left(g(w) - \xi\right).$$

As we shall see below, the proxy PL inequality is a natural extension of the standard PL inequality.

Our main contributions are as follows.

1. When $f$ satisfies $(g, h)$-proxy convexity, and $f$ is either Lipschitz or satisfies a particular smoothness assumption, then for any norm bound $R > 0$, the online SGD algorithm run for polynomial (in $1/\varepsilon$ and $R$) number of iterations satisfies the following in expectation over $z_1, \ldots, z_T \sim \mathcal{D}^T$,

$$\min_{t<T} \mathbb{E}_{z\sim\mathcal{D}} g(w_t; z) \leq \min_{\|w\|\leq R} \mathbb{E}_{z\sim\mathcal{D}} h(w; z) + \varepsilon.$$

2. When $f$ satisfies a $(g, \xi, \alpha, \mu)$-proxy PL inequality and has Lipschitz gradients, SGD run for a polynomial (in $1/\varepsilon$) number of iterations satisfies the following in expectation over $z_1, \ldots, z_T \sim \mathcal{D}^T$,

$$\min_{t<T} \mathbb{E}_{z\sim\mathcal{D}} g(w_t; z) \leq \xi + \varepsilon.$$

3. We demonstrate that many previous guarantees for neural networks trained by gradient descent can be unified in the framework of proxy convexity.

As we will describe in more detail below, if a loss function $\ell$ is $(g, h)$-proxy convex or satisfies a $g$-proxy PL inequality, then the optimization problem is straightforward and the crux of the problem then becomes connecting guarantees for the proxy $g$ with approximate guarantees for $f$.

**Notation.** We use uppercase letters to refer to matrices, and lowercase letters will either refer to vectors or scalars depending on the context. For vectors $w$, we use $\|w\|$ to refer to the Euclidean norm, and for matrices $W$ we use $\|W\|$ to refer to the Frobenius norm. We use the standard $O(\cdot)$, $\Omega(\cdot)$ notations to hide universal constants, with $\tilde{O}(\cdot)$ and $\tilde{\Omega}(\cdot)$ additionally hiding logarithmic factors.

## 2 Proxy Convexity in Comparison to Other Non-convex Optimization Frameworks

In this section, we describe how proxy convexity and proxy PL-inequalities relate to other notions in non-convex optimization. In Section 6, we will discuss additional related work. First, recall that a function $f$ is $(g, h)$-proxy convex if there exist functions $g$ and $h$ such that for all $w, v$,

$$\langle \nabla f(w), w - v \rangle \geq g(w) - h(v).$$

One notion from the non-convex optimization literature that is related to our notion of proxy convexity is that of *invexity* [19]. A function $f$ is invex if it is differentiable and there exists a vector-valued function $k(w, v)$ such that for any $w, v$,

$$\langle \nabla f(w), k(w, v) \rangle \geq f(w) - f(v).$$

It has been shown that a smooth function $f$ is invex if and only if every stationary point of $f$ is a global minimum [10]. However, for many problems of interest involving neural networks, it is not the case that every stationary point will be a global optimum, which makes invexity a less appealing framework for understanding neural networks. Indeed, we shall see in Example 4.2 below that if one considers the problem of learning a single ReLU neuron $x \mapsto \max(0, \langle w, x \rangle)$ under the squared loss, it is not hard to see that there exist stationary points which are not global minima (e.g., $w = 0$ assuming the convention $\sigma'(0) = 0$). By contrast, we shall see that the single ReLU neuron does satisfy a form of proxy convexity that enables SGD to find approximately (but not globally) optimal minima. Thus even the simplest neural networks induce objective functions which are proxy convex and non-invex. We shall see in Example 3.3 that proxy convexity appears in the objective functions induced by wide and deep neural networks as well.

To understand how the proxy PL inequality compares to other notions in the optimization literature, recall that an objective function $f$ satisfies the standard PL inequality [33, 27] if there exists $\mu > 0$ such that

$$\|\nabla f(w)\|^2 \geq \frac{\mu}{2} [f(w) - f^*] \quad \forall w,$$

where $f^* = \min_w f(w)$. Clearly, any stationary point of an objective satisfying the standard PL inequality is globally optimal. Thus, the presence of local minima among stationary points in neural network objectives makes the standard PL inequality suffer from the same drawbacks that invexity does for understanding neural networks. This further applies to any of the conditions which are known to imply the PL inequality, like weak strong convexity, the restricted secant inequality, and the error bound condition (Karimi et al. [23]).[1]

In comparison, the $(g, \xi, \alpha, \mu)$-proxy PL inequality is satisfied if there exists a function $g$ and constants $\xi > 0$, $\alpha \in (0, 2]$ and $\mu > 0$ such that

$$\|\nabla f(w)\|^\alpha \geq \frac{\mu}{2} [g(w) - \xi] \quad \forall w.$$

It is clear that if a function $f$ satisfies the standard PL inequality, then it satisfies the $(f, f^*, 2, \mu)$ proxy PL inequality. Stationary points $w_0$ of objective functions satisfying the proxy PL inequality have $\|\nabla f(w_0)\| = 0$ which imply $g(w_0) \leq \xi$. In the case that $g = f$, the slack error term $\xi$ allows for the proxy PL inequality framework to accommodate the possibility that stationary points may not be globally optimal (i.e. have objective value $f^* = \min_w f(w)$), but could be approximately optimal by, for example, having objective value at most $\xi = C \cdot f^*$ or $\xi = C \cdot \sqrt{f^*}$ for some constant $C \geq 1$. When $g \neq f$, the proxy PL inequality allows for the possibility of analyzing a proxy loss function $g$ which is *implicitly* minimized when using gradient-based optimization of the objective $f$.

At a high level, proxy convexity and the proxy PL inequality are well-suited to situations where stationary points may not be *globally* optimal, but may be approximately optimal with respect to a related optimization objective. The proxy convexity framework allows for one to realize this through developing problem-specific analyses that connect the proxy objective $g$ to the original objective $f$. As we shall see below, rich function classes like neural networks are often more easily analyzed by considering a proxy objective function that naturally appears when one analyzes the gradient of the loss.

Finally, we note that [25] introduced a different generalization of the PL inequality, namely the $\text{PL}^*$ and $\text{PL}^*_\varepsilon$ inequality, which relaxes the standard PL inequality definition so that the PL condition only needs to hold on a subset of the domain. In particular, a function $f : \mathbb{R}^p \to \mathbb{R}$ satisfies the $\text{PL}^*$ inequality on a set $S \subset \mathbb{R}^p$ if there exists $\mu > 0$ such that

$$\|\nabla f(w)\|^2 \geq \mu f(w) \quad \forall w \in S.$$

Likewise, $f$ satisfies the $\text{PL}^*_\varepsilon$ inequality on $S$ if there exists a set $S$ and $\varepsilon \geq 0$ such that the $\text{PL}^*$ inequality holds on the set $S_\varepsilon = \{w \in S : f(w) \geq \varepsilon\}$. One can see that if $f$ satisfies the $\text{PL}^*_\varepsilon$ inequality on $S$, then the function $g(w) := f(w) + \varepsilon$ satisfies the $g$-proxy, $\varepsilon$-optimal PL inequality on $S$.

We wish to emphasize the differences in the framing and motivation of the $\text{PL}^*_\varepsilon$ inequality by [25] and that of proxy convexity and the proxy PL inequality in this paper. [25] focus on the geometry of

---

[1]Karimi et al. [23] shows that these conditions imply the PL inequality under the assumption that the objective function has Lipschitz-continuous gradients.

optimization in the overparameterized setting where one has a fixed set of samples $\{(x_i, y_i)\}_{i=1}^n$ and a parametric model class $g(x; w)$ (for $w \in \mathbb{R}^p$, $p > n$) and the goal is to solve $g(x_i; w) = y_i$ for all $i \in [n]$. In this setting one can view the optimization problem as a nonlinear least squares system with $p$ unknowns and $n$ equations, and [25] use geometric arguments to show that when $p > n$ the PL$^*$ condition is satisfied throughout most of the domain. They extend the PL$^*$ condition to the PL$^*_\varepsilon$ condition with the motivation that in underparameterized settings, or when performing early stopping, there may not exist interpolating solutions. By contrast, we focus on the stochastic optimization setting, where the goal is to minimize the expected loss over some distribution; it is unclear how geometric arguments for minimizing the training loss can lead to generalization guarantees in the overparameterized setting, especially when considering online SGD where samples are observed one-by-one and 'overparameterization' has a less clear meaning. Furthermore, in this work we are not primarily interested in understanding how overparameterization affects optimization. Rather, our aim is to develop a framework that allows for formal characterizations of optimization problems where stationary points are not globally optimal with respect to the original objective but are approximately optimal with respect to proxy objective functions. We will demonstrate below that such a framework can help unify a number of works on learning with neural networks trained by gradient descent.

## 3 Proxy PL Inequality Implies Proxy Objective Guarantees

In this section, we show that for loss functions satisfying a proxy PL inequality, SGD[2] efficiently minimizes the proxy. We leave the proofs for Section 5.

**Theorem 3.1.** *Suppose $F(w) = \mathbb{E}_{z \sim \mathcal{D}} f(w; z)$ where $f(\cdot; z)$ satisfies the $(g(\cdot; z), \xi(z), \alpha, \mu)$-proxy PL inequality for some function $g(\cdot; z) : \mathbb{R}^p \to \mathbb{R}$ for each $z$. Denote by $G(w) := \mathbb{E}_{z \sim \mathcal{D}} g(w; z)$. Assume that $f$ is non-negative and has $L_2$-Lipschitz gradients. Then for any $\varepsilon > 0$, provided $\eta < 1/L_2$, online SGD with fixed step size $\eta$ and run for $T = 2\eta^{-1}(\mu \varepsilon / 2)^{-2/\alpha} f(w_0; z_0)$ iterations results in the following guarantee in expectation over $z_0, \ldots, z_{T-1} \sim \mathcal{D}^T$,*

$$\min_{t < T} G(w_t) \leq \mathbb{E}_{z \sim \mathcal{D}} \xi(z) + \varepsilon. \tag{3.1}$$

To get a feel for how a proxy PL inequality might be useful for learning neural networks, consider a classification problem with labels $y \in \{\pm 1\}$, and suppose that $N(W; x)$ is a neural network function. A standard approach for learning neural networks is to minimize the cross-entropy loss $\ell(yN(W; x)) = \log(1 + \exp(-yN(W; x)))$ using gradient descent. Using the variational form of the norm, we have

$$\|\nabla \ell(yN(W; x))\| = \sup_{\|U\|=1} \langle \nabla \ell(yN(W; x)), U \rangle$$
$$\geq -\ell'(yN(W; x)) \cdot y \langle \nabla N(W; x), V \rangle, \tag{3.2}$$

where $V$ is any matrix satisfying $\|V\| = 1$. Now, although the function $-\ell'$ is not an upper bound for $\ell$ (indeed, $-\ell' < \ell$), it *is* an upper bound for a constant multiple of the zero-one loss, and can thus serve as a proxy for the classification error.[3] This is because for convex and decreasing losses $\ell$, the function $-\ell'$ is non-negative and decreasing, and hence by Markov's inequality,

$$\mathbb{P}(y \neq \text{sgn}(N(W; x))) = \mathbb{P}(y \cdot N(W; x) < 0)$$
$$= \mathbb{P}(-\ell'(yN(w; x)) > -\ell'(0))$$
$$\leq \frac{1}{-\ell'(0)} \mathbb{E}[-\ell'(yN(W; x))].$$

Thus, if one can bound the population risk under $-\ell'$, one has a bound for the classification error. Indeed, this property has been used in a number of recent works on neural networks [6, 13, 22, 16]. This lets the $-\ell'$ term in (3.2) represent the desired proxy $g$ in the definition of the $(g, \xi, \alpha, \mu)$-proxy PL inequality. Thus, for neural network classification problems, the problem of showing the neural network has small classification error is reduced to constructing a matrix $V$ that allows for the quantity $y \langle \nabla N(W; x), V \rangle$ to be large and non-negative. The quantity $y \langle \nabla N(W; x), V \rangle$ can be thought of

---

[2]We focus on online SGD in this paper for simplicity. Analogous optimization guarantees would hold for other variants of gradient descent that utilize samples in batches.

[3]In fact, the function $z \mapsto [\ell'(z)]^2$ is also a proxy for the classification error; see [15, Appendix A].

as a margin function that is large when the gradient of the neural network loss points in a good direction. Although we shall see below that in some instances one can derive a lower bound for $y\langle\nabla N(W;x),V\rangle$ that holds for *all* $W$, $x$, and $y$, a more general approach would be to show that along the gradient descent trajectory $W^{(t)}$, a lower bound for $y\langle\nabla N(W^{(t)};x),V\rangle$ holds.[4]

In the remainder of this section, we will show that a number of recent works on learning neural networks with gradient descent utilized proxy PL inequalities. In our first example, we consider recent work by Charles and Papailiopoulos [7] that directly used a (standard) PL inequality.

**Example 3.2** (Standard PL inequality for single leaky ReLU neurons and deep linear networks)**.** Charles and Papailiopoulos [7] showed that the standard PL inequality holds in two distinct settings. The first is that of a single leaky ReLU neuron $x \mapsto \sigma(\langle w,x\rangle)$, where $\sigma(z) = \max(c_\sigma z, z)$ for $c_\sigma \neq 0$. They showed that if $s_{\min}(X)$ is the smallest singular value of the matrix $X \in \mathbb{R}^{n\times d}$ of $n$ samples, then for a $\lambda$-strongly convex loss $\ell$, the loss $f(w) = \ell(\sigma(\langle w,x\rangle))$ satisfies the standard $\mu$-PL inequality, i.e., the $(f, f^*, 2, \mu)$-proxy PL inequality for $\mu = \lambda s_{\min}(X)^2 c_\sigma^2$ (Charles and Papailiopoulos [7, Theorem 4.1]).

The same authors also showed that under certain conditions the standard PL inequality holds when the neural network takes the form $N(W;x) = W_L \cdots W_1 x$ and the loss is the squared loss, $f(W) = \frac{1}{2}(y - N(W;x))^2$. In particular, they showed that if $s_{\min}(W_i) \geq \tau > 0$ throughout the gradient descent trajectory, then $f$ satisfies the standard $\mu$-PL inequality for $\mu = L\tau^{2L-2} / \left\|(XX^\top)^{-1}X\right\|_F^2$ (Charles and Papailiopoulos [7, Theorem 4.5]).

The standard PL inequality has been used by a number of other authors in the deep learning theory literature, see e.g. Xie et al. [36, Theorem 1], Hardt and Ma [20, Eq. 2.3], Zhou and Liang [40, Theorem 1], Shamir [35, Theorem 3].

In our next example, we show that a proxy PL inequality holds for deep neural networks in the neural tangent kernel (NTK) regime.

**Example 3.3** (Proxy PL inequality for deep neural networks in NTK regime)**.** Consider the class of deep, $L$-hidden-layer ReLU networks, either with or without residual connections:

$$N_1(W;x) = \sigma(W^{(1)}x), \quad N_l(W;x) = s_l N_{l-1}(W;x) + \sigma(W^{(l)} N_{l-1}(W;x)), \, l = 2,\ldots,L,$$

$$N(W;x) = \sum_{j=1}^m a_j [N_L(W;x)]_j,$$

where $s_l = 0$ for fully-connected networks and $s_l = 1$ for residual networks. Cao and Gu [6, Theorem 4.2], Frei, Cao, and Gu [13, Lemma 4.3], and Zou et al. [41, Lemma B.5] have shown that under certain distributional assumptions and provided the iterates of gradient descent stay close to their intialization, one can guarantee that for the cross-entropy loss $f(W;(x,y)) = \ell(yN(W;x))$,

$$\|\nabla f(W;(x,y))\| \geq C_1 \cdot -\ell'(yN(W;x)). \tag{3.3}$$

By defining $g(W;z) = -\ell'(yN(W;x))$, the loss $f$ satisfies the $(g, 0, 1, 2C_1)$-proxy PL inequality. Since the ReLU is not smooth, the loss $f$ will not have Lipschitz gradients, and thus a direct application of Theorem 3.1 is not possible. Instead, the authors show that in the NTK regime, the loss obeys a type of semi-smoothness that still allows for an analysis simliar to that of Theorem 3.1.

**Example 3.4** (Proxy PL inequality for one-hidden-layer networks outside NTK regime)**.** Consider a one-hidden-layer network with activation function $\sigma$,

$$N(W;(x,y)) = \sum_{j=1}^m a_j \sigma(\langle w_j, x\rangle), \tag{3.4}$$

where the second layer weights $\{a_j\}_{j=1}^m$ are randomly initialized and fixed at initialization, but the $\{w_j\}_{j=1}^m$ are trained. Assume $\sigma$ satisfies $\sigma'(z) \geq c_\sigma > 0$ for all $z$ (e.g., the leaky ReLU activation). Frei, Cao, and Gu have shown [16, Lemma 3.1] that there exists a matrix $V \in \mathbb{R}^{m\times d}$ with $\|V\|_F = 1$ such that for distributions satisfying anti-concentration, for any $x, y$ and $W$,

$$y\langle\nabla N(W;x),V\rangle \geq C_1[c_\sigma - \xi(x,y)],$$

---

[4]Although our results as stated would not immediately apply in this setting, the proof would be the same up to trivial modifications.

where $\mathbb{E}[\xi(x,y)] = O(\sqrt{\mathsf{OPT}})$ where $\mathsf{OPT}$ is the best classification error achieved by a halfspace over $\mathcal{D}$. Thus, when $f(W; (x,y)) = \ell(yN(W; (x,y)))$ is the cross-entropy loss,

$$
\begin{aligned}
\|\nabla f(W; (x,y))\| &= \sup_{\|Z\|_F=1} \langle \nabla f(W; (x,y)), Z \rangle \\
&\geq -\ell'(yN(W; x)) \cdot y \langle \nabla N(W; x), V \rangle \\
&\geq C_1 c_\sigma \cdot [-\ell'(yN(W; x)) - c_\sigma^{-1}\xi(x,y)].
\end{aligned}
$$

As in Example 3.3, by defining $g(W; z) = -\ell'(yN(W; x))$, the loss $f$ satisfies the $(g, c_\sigma^{-1}\xi(x,y), 1, 2C_1 c_\sigma)$-proxy PL inequality. Thus, provided we can show that $f(\cdot; z)$ has $L_2$-Lipschitz gradients for some constant $L_2 > 0$, Theorem 3.1 shows that

$$
\begin{aligned}
\min_{t<T} \mathbb{P}_{(x,y)}(y \neq \mathrm{sgn}(N(W^{(t)}; x))) &\leq \min_{t<T} \frac{\mathbb{E}_{(x,y)\sim\mathcal{D}}[-\ell'(yN(W^{(t)}; x)]}{|\ell'(0)|} \\
&\leq 2C_2 c_\sigma^{-1}\mathbb{E}_{(x,y)\sim\mathcal{D}}\xi(x,y) + \varepsilon \\
&= O(\sqrt{\mathsf{OPT}}) + \varepsilon.
\end{aligned}
$$

Provided $\sigma$ is such that $\sigma'$ is continuous and differentiable, then $f$ has $L_2$-Lipschitz gradients and thus the guarantees will follow. In particular, this analysis follows if $\sigma$ is any smoothed approximation to the leaky ReLU which satisfies $\sigma'(z) \geq c_\sigma > 0$.

Note that the above optimization analysis is an original contribution of this work as we utilize a completely different proof technique than that of [16]. In that paper, the authors utilize a Perceptron-style proof technique that analyzes the correlation $\langle W^{(t)}, V \rangle$ of the weights found by gradient descent and a reference matrix $V$. Their proof relies crucially on the homogeneity of the (non-smooth) leaky ReLU activation, namely that $z\sigma'(z) = \sigma(z)$ for $z \in \mathbb{R}$, and cannot accommodate more general smooth activations. By contrast, the proxy PL inequality proof technique in this example relies upon the smoothness of the activation function and is more similar to smoothness-based analyses of gradient descent.

# 4 Proxy Convexity Implies Proxy Objective Guarantees

In this section, we show that if $f$ satisfies $(g, h)$-proxy convexity, we can guarantee that by minimizing $f$ with gradient descent, we find a hypothesis for which $g(w)$ is at least as small as the best norm-bounded predictor as measured by the loss $h$. We present two versions of our result: one that relies upon fewer assumptions on the loss $f$ but needs a small step size, and another that requires a proxy smoothness assumption on $f$ but allows for a constant step size. The proofs for the theorem are given in Section 5.

**Theorem 4.1.** *Suppose that $F(w) := \mathbb{E}_{z\sim\mathcal{D}}f(w; z)$ and $f(\cdot; z)$ is $(g(\cdot; z), h(\cdot; z))$-proxy convex for each $z$. Denote $H(w) := \mathbb{E}_{z\sim\mathcal{D}}h(w; z)$ and $G(w) := \mathbb{E}_{z\sim\mathcal{D}}g(w; z)$.*

*(a) Assume there exists $L_1 > 0$ such that for all $w$, $\mathbb{E}_{z\sim\mathcal{D}}[\|\nabla f(w; z)\|^2] \leq L_1^2$. Then for any $v \in \mathbb{R}^p$ and any $\varepsilon > 0$, performing online SGD on $F(w)$ from an arbitrary initialization $w_0$ with fixed step size $\eta \leq \varepsilon L_1^{-2}$ for $T = \eta^{-1}\varepsilon^{-1}\|w_0 - v\|^2$ iterations implies that, in expectation over $(z_0, \ldots, z_{T-1}) \sim \mathcal{D}^T$,*

$$\min_{t<T} G(w_t) \leq H(v) + \varepsilon.$$

*(b) Assume there exists $L_2 > 0$ such that for all $w$, $\mathbb{E}_{z\sim\mathcal{D}}[\|\nabla f(w; z)\|^2] \leq 2L_2\mathbb{E}_{z\sim\mathcal{D}}g(w; z)$. Then for any $v \in \mathbb{R}^p$ and any $\varepsilon > 0$, performing online SGD on $F(w)$ from an arbitrary initialization with fixed step size $\eta \leq L_2^{-1}/2$ for $T = \eta^{-1}\varepsilon^{-1}\|w_0 - v\|^2$ implies that, in expectation over $(z_0, \ldots, z_{T-1}) \sim \mathcal{D}^T$,*

$$\min_{t<T} G(w_t) \leq (1 + 2\eta L_2)H(v) + \varepsilon.$$

In order for $(g, h)$-proxy convexity to be useful, there must be a way to relate guarantees for $g$ into guarantees for the desired objective function $f$. In the remainder of this section, we will discuss two neural network learning problems which are non-convex and yet satisfy proxy convexity which leads to generalization guarantees. Our first example is the problem of learning a neural network with a single nonlinear unit.

**Example 4.2** (Single neuron satisfies proxy convexity). Consider the problem of learning a single neuron $x \mapsto \sigma(\langle w, x \rangle)$ under the squared loss, where $\sigma$ is the ReLU activation. The objective function of interest is

$$F(w) = \mathbb{E}_{(x,y) \sim \mathcal{D}} 1/2 (\sigma(\langle w, x \rangle) - y)^2.$$

Denote

$$F^* := \min_{\|w\| \leq 1} F(w).$$

It is known that $F$ is non-convex [37]. Under the assumption that learning sparse parities with noise is computationally hard, it is known that no polynomial time algorithm can achieve risk $F^*$ exactly; moreover, it is known that (unconditionally) the standard gradient descent algorithm cannot achieve risk $F^*$ [18].[5] However, Frei, Cao, and Gu [14] showed that although $F$ is non-convex and no algorithm can achieve risk $F^*$, $F$ does satisfy a form of proxy convexity that allows for gradient descent to achieve risk $O(\sqrt{F^*})$. They showed that the loss function

$$f(w; (x, y)) = 1/2 (\sigma(\langle w, x \rangle) - y)^2$$

satisfies $(g, h)$-proxy convexity, where

$$g(w; (x, y)) = 2 \left[ \sigma(\langle w, x \rangle) - \sigma(\langle v^*, x \rangle) \right]^2 \sigma'(\langle w, x \rangle),$$
$$h(v; (x, y)) = |\sigma(\langle v, x \rangle) - y| = \sqrt{2f(v; (x, y))},$$

where $v^*$ is the population risk minimizer of $F(w)$ (see their Eq. (3.13)). Moreover, they showed (see their Eq. (3.9))

$$\|\nabla f(w; z)\|^2 \leq 8g(w; z).$$

Thus Theorem 4.1(b) implies that SGD with step size $\eta \leq 1/8$ and $T = 2\eta^{-1}\varepsilon^{-1} \|w_0 - v^*\|^2$ iterations will find a point $w_t$ satisfying

$$\begin{aligned}
G(w_t) = 2\mathbb{E}_{(x,y)} &\left[ (\sigma(\langle w_t, x \rangle) - \sigma(\langle v^*, x \rangle))^2 \sigma'(\langle w_t, x \rangle) \right] \\
&\leq (1 + 8\eta) H(v^*) + \varepsilon \\
&\leq (1 + 8\eta) \mathbb{E} |\sigma(\langle v^*, x \rangle) - y| + \varepsilon \\
&\leq (1 + 8\eta) \sqrt{\mathbb{E}[(\sigma(\langle v^*, x \rangle) - y)^2]} + \varepsilon \\
&= O(\sqrt{F^*}).
\end{aligned}$$

The authors then show that under some distributional assumptions on $\mathcal{D}$, $G(w_t) = O(\sqrt{F^*})$ implies $F(w_t) = O(\sqrt{F^*})$ [14, Lemma 3.5]. Thus, the optimization problem for $F$ induces a proxy convex optimization problem defined in terms of $G$ which yields guarantees for $G$ in terms of $H$, and this in turn leads to approximate optimality guarantees for the original objective $F$.

In our next example, we show that a number of works on learning one-hidden-layer ReLU networks in the neural tangent kernel regime [21] can be cast as problems satisfying proxy convexity.

**Example 4.3** (Proxy convexity for one-hidden-layer ReLU neural networks in the NTK regime). Consider the class of one-hidden-layer ReLU networks consisting of $m$ neurons,

$$N(W; (x, y)) = \sum_{j=1}^{m} a_j \sigma(\langle w_j, x \rangle),$$

where the $\{a_j\}_{j=1}^m$ are randomly initialized and fixed at initialization, but the $\{w_j\}_{j=1}^m$ are trained. Suppose we consider a binary classification problem, where $y \in \{\pm 1\}$ and we minimize the cross-entropy loss,

$$F(W) = \mathbb{E}_{(x,y) \sim \mathcal{D}} f(W; (x, y)), \quad f(W; (x, y)) = \ell(yN(W; (x, y))), \quad \ell(z) = \log(1 + \exp(-z)).$$

Ji and Telgarsky [22, Proof of Lemma 2.6] showed that there exists a function $\tilde{h}(a, W, V; (x, y))$ such that the iterates of gradient descent satisfy

$$\langle \nabla f(W; (x, y)), w - v \rangle \geq f(W; (x, y)) - \tilde{h}(a, W, V; (x, y)).$$

---

[5]This stands in contrast to learning a single *leaky* ReLU neuron $x \mapsto \max(\alpha x, x)$ for $\alpha \neq 0$, which as we showed in Example 3.2 can be solved using much simpler techniques.

Under the assumption that the iterates of gradient descent stay close to the initialization (i.e., the neural tangent kernel regime), they show that $\tilde{h}(a, W, V; (x, y)) \leq \varepsilon$ under distributional assumptions, and thus $F(w)$ will satisfy $(f, \tilde{h} \equiv \varepsilon)$-proxy convexity. The cross entropy loss satisfies $[\ell'(z)]^2 \leq \ell(z)$, and thus we can apply Theorem 4.1(b) to get guarantees of the form $\min_{t<T} F(w_t) \leq \varepsilon$ for the cross-entropy loss $F(W)$ of SGD-trained neural networks in the NTK regime.

In another problem of learning one-hidden-layer networks, Allen-Zhu, Li, and Liang [3, Proof of Lemma B.4] show that there exists a proxy loss function $g(a, W; (x, y))$ such that provided the neural network weights stay close to their initialized values, $f(a, W; (x, y))$ satisfies $(g, g + \varepsilon)$ proxy convexity. Again using that the cross-entropy loss satisfies $[\ell'(z)]^2 \leq \ell(z)$, Theorem 4.1(b) shows that SGD-trained neural networks in the NTK regime satisfy $\min_{t<T} G(W_t) \leq \min_V G(V) + \varepsilon$. They further show that the proxy loss $g$ is close to the cross entropy loss, i.e. $|\mathbb{E}[g(a, W; (x, y))] - \mathbb{E}[f(a, W; (x, y))]| \leq \varepsilon$, implying a bound of the form $\min_{t<T} F(W_t) \leq \min_V F(V) + \varepsilon$.

## 5 Proof of the Main Results

In this section we provide the proofs of the theorems given in Sections 3 and 4.

We first give the proof of Theorem 3.1 which provides guarantees for learning with objectives satisfying proxy PL inequalities.

*Proof of Theorem 3.1.* Since $f$ has $L_2$-Lipschitz gradients, we have for any $w, w'$ and fixed $z$,

$$f(w; z) \leq f(w'; z) + \langle \nabla f(w'; z), w - w' \rangle + \frac{L_2}{2} \|w - w'\|^2.$$

Taking $w = w_{t+1}$, $w' = w_t$, and $z = z_t$,

$$f(w_{t+1}; z_t) \leq f(w_t; z_t) - \eta \|\nabla f(w_t; z_t)\|^2 + \frac{\eta^2 L_2}{2} \|\nabla f(w_t; z_t)\|^2$$
$$= f(w_t; z_t) - \eta [1 - \eta L_2/2] \|\nabla f(w_t; z_t)\|^2. \tag{5.1}$$

Since $\eta < 1/L_2$, we have $(1 - \eta L_2/2)^{-1} \leq 2$, and thus we can rearrange the above to get

$$\|\nabla f(w_t; z_t)\|^2 \leq \frac{1}{\eta(1 - \eta L_2/2)}[f(w_t; z_t) - f(w_{t+1}; z_t)]$$
$$\leq \frac{2}{\eta}[f(w_t; z_t) - f(w_{t+1}; z_t)]. \tag{5.2}$$

Summing the above from $t = 0$ to $t = T - 1$ and using that $f$ is non-negative, we get

$$\frac{1}{T} \sum_{t=0}^{T-1} \|\nabla f(w_t; z_t)\|^2 \leq \frac{2f(w_0; z_0)}{\eta T}.$$

Using the definition of proxy PL inequality, this implies

$$\frac{1}{T} \sum_{t=0}^{T-1} (\mu/2)^{2/\alpha} (g(w_t; z_t) - \xi(z_t))^{2/\alpha} \leq \frac{2f(w_0; z_0)}{\eta T}.$$

Taking the minimum over $t < T$ and re-arranging terms, this means

$$\min_{t<T} (g(w_t; z_t) - \xi(z_t))^{2/\alpha} \leq \frac{2f(w_0; z_0)}{\eta T (\mu/2)^{2/\alpha}}.$$

Therefore, we have

$$\min_{t<T} g(w_t; z_t) \leq \xi(z_t) + \frac{2}{\mu} \cdot \left( \frac{2f(w_0; z_0)}{\eta T} \right)^{\alpha/2}.$$

Taking $T = 2\eta^{-1} f(w_0; z_0)(\mu \varepsilon/2)^{-2/\alpha}$ and taking expectations over $z_0, \ldots, z_{T-1}$, we get (3.1). □

We next prove guarantees for SGD when the objective satisfies proxy convexity.

*Proof of Theorem 4.1.* By the definition of proxy convexity,

$$\|w_t - v\|^2 - \|w_{t+1} - v\|^2 = 2\eta\langle\nabla f(w_t; z_t), w_t - v\rangle - \eta^2 \|\nabla f(w_t; z_t)\|^2$$

$$\geq 2\eta[g(w_t; z_t) - h(v; z_t)] - \eta^2 \|\nabla f(w_t; z_t)\|^2.$$

Conditional on the values of $z_0, \ldots, z_{t-1}$ (and hence on the value of $w_t$), taking expectations of both sides with respect to $z_t \sim \mathcal{D}$ results in

$$\|w_t - v\|^2 - \mathbb{E}_{z_t \sim \mathcal{D}}\|w_{t+1} - v\|^2 \geq 2\eta[G(w_t) - H(v)] - \eta^2 \mathbb{E}_{z_t \sim \mathcal{D}}\|\nabla f(w_t; z_t)\|^2. \qquad (5.3)$$

For case (a), this results in

$$\|w_t - v\|^2 - \mathbb{E}_{z_t \sim \mathcal{D}}\|w_{t+1} - v\|^2 \geq 2\eta[G(w_t) - H(v) - \eta/2L_1^2].$$

Dividing both sides by $2\eta T$ and summing from $t = 0, \ldots, T-1$, we get

$$\frac{1}{T}\sum_{t=0}^{T-1} G(w_t) \leq \frac{1}{T}\sum_{t=0}^{T-1} H(v) + \frac{\eta L_1^2}{2} + \frac{\|w_0 - v\|^2 - \mathbb{E}_{z_{T-1} \sim \mathcal{D}}\|w_T - v\|^2}{2\eta T}.$$

Taking expectations over $z_{0:t} = (z_0, \ldots, z_{t-1}) \sim \mathcal{D}^t$, we get

$$\min_{t < T} \mathbb{E}_{z_{0:t} \sim \mathcal{D}^t} G(w_t) \leq \frac{1}{T}\sum_{t=0}^{T-1}\mathbb{E}_{z_{0:t} \sim \mathcal{D}^t} G(w_t) \leq H(v) + \frac{\eta L_1^2}{2} + \frac{\|w_0 - v\|^2}{2\eta T}.$$

In particular, for $\eta \leq \varepsilon L_1^{-2}$ and $T = \eta^{-1}\varepsilon^{-1}\|w_0 - v\|^2$, we get

$$\min_{t < T} \mathbb{E}_{z_{0:t} \sim \mathcal{D}^t} G(w_t) \leq H(v) + \varepsilon.$$

For case (b), (5.3) becomes

$$\|w_t - v\|^2 - \mathbb{E}_{z_t \sim \mathcal{D}}\|w_{t+1} - v\|^2 \geq 2\eta[G(w_t) - H(v) - \eta L_2 G(w_t)]$$

$$= 2\eta\left[(1 - \eta L_2)G(w_t) - H(v)\right].$$

Dividing both sides by $2\eta T(1 - \eta L_2)$ and summing from $t = 0, \ldots, T-1$, we get

$$\frac{1}{T}\sum_{t=0}^{T-1} G(w_t) \leq \frac{1}{1 - \eta L_2} H(v) + \frac{\|w_0 - v\|^2 - \mathbb{E}_{z_{T-1} \sim \mathcal{D}}\|w_T - v\|^2}{2\eta T(1 - \eta L_2)}$$

$$\leq (1 + 2\eta L_2)H(v) + \frac{(1 + 2\eta L_2)\|w_0 - v\|^2}{2\eta T},$$

where in the last line we have used that $\eta \leq L_2^{-1}/2$ and that $1/(1-x) \leq 1 + 2x$ on $[0, 1/2]$. Taking expectations over $z_{0:t} = (z_0, \ldots, z_{t-1}) \sim \mathcal{D}^t$, we get

$$\min_{t < T} \mathbb{E}_{z_{0:t} \sim \mathcal{D}^t} G(w_t) \leq \frac{1}{T}\sum_{t=0}^{T-1}\mathbb{E}_{z_{0:t} \sim \mathcal{D}^t} G(w_t)$$

$$\leq (1 + 2\eta L_2)H(v) + \frac{2\|w_0 - v\|^2}{2\eta T}.$$

In particular, for $T = \eta^{-1}\varepsilon^{-1}\|w_0 - v\|^2$, we get

$$\min_{t < T} \mathbb{E}_{z_{0:t} \sim \mathcal{D}^t} G(w_t) \leq (1 + 2\eta L_2)H(v) + \varepsilon.$$

$\square$

We note that under additional assumptions on $\mathcal{D}$ and the loss function, we could improve the results from holding in expectation over the draws of the sample to high probability guarantees by using standard concentration arguments. This is easily done when the objective function satisfies proxy convexity: we can make a slight modification to the proof of Theorem 4.1 to argue inductively that until we reach a point with $G(w_t) \leq H(v) + \eta L_1^2 + \varepsilon$, we have that that $\|w_t - v\|^2 - \|w_{t+1} - v\|^2 \geq \varepsilon$. This implies that the norm of the predictors remain bounded throughout the trajectory of gradient descent until we reach the desired point with $G(w_t) \leq H(v) + \eta L_1^2 + \varepsilon$, which can then be used in Rademacher complexity-type arguments [4]. This type of argument was previously used by e.g., Frei et al. [14].

# 6 Additional Related Work

The Polyak–Lojasiewicz inequality can be dated back to the original works of Polyak [33] and Lojasiewicz [27]. Recent work by Karimi et al. [23] proved linear convergence under the PL condition and showed that the PL condition is one of the weakest assumptions under which linear convergence is possible. In particular, they showed that the error bound inequality [28], essential strong convexity [26], weak strong convexity [31], and the restricted secant inequality [38] are all assumptions under which linear convergence is possible and that each of these assumptions implies the PL inequality.

As we described in Section 2, the standard PL condition was shown to hold under certain assumptions for neural network objective functions [20, 36, 40, 7]. In addition to those covered in this paper, there are a number of other provable guarantees for generalization of SGD-trained networks which rely on a variety of different techniques, such as tensor methods [24] and utilizing connections with partial differential equations by way of mean field approximations [29, 9, 30, 8].

In the optimization literature, recent work has shown that SGD can efficiently find stationary points and can escape saddle points [17, 12]. As the proxy PL inequality implies guarantees for the proxy objective function at stationary points of the original optimization objective, our framework can naturally be used for other optimization algorithms that are known to efficiently find stationary points, such as SVRG [2, 34], Natasha2 [1], SARAH/SPIDER [32, 11], and SNVRG [39].

# 7 Conclusion

In this paper we have introduced the notion of proxy convexity and proxy PL inequality and developed guarantees for learning with stochastic gradient descent under these conditions. We demonstrated that many recent works in the learning of neural networks with gradient descent can be framed in terms of optimization problems that satisfy either proxy convexity or a proxy PL inequality. While the proxy convexity framework cannot unify all existing analyses of learning neural networks, we hope that it can reveal some of the principles underlying the success of SGD-trained neural networks.

## Acknowledgments and Disclosure of Funding

QG is partially supported by the National Science Foundation CAREER Award 1906169, IIS-1855099 and IIS-2008981. SF acknowledges the support of the NSF and the Simons Foundation for the Collaboration on the Theoretical Foundations of Deep Learning through awards DMS-2031883 and #814639. The views and conclusions contained in this paper are those of the authors and should not be interpreted as representing any funding agencies.

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
