# OpenReview forum: "Proxy Convexity: A Unified Framework for the Analysis of Neural Networks Trained by Gradient Descent"
_NeurIPS.cc/2021/Conference — NeurIPS 2021 Poster_

### Official Review · Reviewer_V96x · 2021-07-01

**Rating:** 7
**Confidence:** 4

**Summary:**

The authors propose the notion of proxy convexity and proxy PL inequalities, which represent whether the original objective function induce a proxy function which is implicitly minimized with SGD. The authors show that many existing convergence guarantees for neural networks trained with SGD can be summarized through this framework of proxy convexity.

**Limitations And Societal Impact:**

Yes.

**Main Review:**

The work in this paper is fundamentally sound and well written, and does a good job of unifying existing results in the literature, such as optimizing a  neural network with a single neuron, or single-layer networks in the NTK regime.

My main concern is with novelty. While the notion of proxy convexity is new and has utility in unifying existing results for SGD applied to neural network optimization, it would nice to have an idea of how this can be useful going forward, such as a new result, or a new setting which these results can be applied to.

**Time Spent Reviewing:**

8

---

> ### Author Response · Authors · 2021-08-10
> **Response to Reviewer V96x**
>
> Thank you for your positive comments.
>
> In response to the question about new settings and new results, as we mention in our response to Reviewer q6Z4, we wish to point out that we do provide a novel optimization analysis in Example 3.4, although it was not clearly detailed as such in the original submission.  In this example we give an alternative proof for Theorem 2.6 of [16, Frei et al 2021].  In that work, the authors relied upon a Perceptron-style proof technique that considers the correlation $\langle W^{(t)}, V\rangle$ between the weights of SGD and those of a fixed weight matrix $V$.  This proof technique relies heavily upon the homogeneity of the leaky ReLU activation (i.e., that $z\sigma'(z)=\sigma(z)$) and would not hold for other activation functions.  By contrast, in Example 3.4 we show that the proxy PL inequality framework enables a guarantee for SGD for networks with smoothed leaky ReLU activations via a more typical smoothness-based optimization analysis.  We emphasize that the Perceptron-style proof of [16] is completely different than the smoothness-based proof we used for the proof of Theorem 3.1 which forms the basis of Example 3.4.   We will be sure to emphasize this contribution in the camera-ready.

---

### Official Review · Reviewer_pCxH · 2021-07-06

**Rating:** 7
**Confidence:** 3

**Summary:**

The authors propose a generalization of the well established notions of convexity and PL inequalities, which they term proxy convexity and proxy PL inequalities correspondingly.  These generalizations are particularly well suited for studying non convex models like neural networks since they do not require all stationary points to be global optima. In fact the authors find that inequalities used in several prior works to prove bounds for neural nets trained with SGD can be unified under proxy convexity and proxy PL inequalities.

**Limitations And Societal Impact:**

Limitations and negative social impact are adequately discussed.

**Main Review:**

To the best of my knowledge,  the authors are the first to study proxy convexity and proxy PL inequalities so their work is definitely novel. While the proofs in Section 5 are not very surprising given the definitions of the proxy inequalities, the main contribution of this work is identifying the unifying pattern in the proof techniques in the literature and its connections with the standard notions of convexity and PL inequalities.

I believe that proxy convexity and proxy PL inequalities can have significant impact. Now that the unifying patterns are observed, it is easier to make the connections between lines of work that study seemingly disparate neural network architectures. Given the generality of the proxy inequalities, follow up work could strive to provide new instantiations of proxy inequalities. Last but not least, identifying powerful proxies that are implicitly minimized by SGD can lead to new loss functions or optimization procedures that minimize the proxies directly.

Overall a strong and clearly written submission. I acknowledge the authors' responses to all reviewers. I believe highlighting the novel analysis of Example 3.4 will further improve the paper. Thus, I remain in favor of accepting this submission.

**Time Spent Reviewing:**

2

---

> ### Author Response · Authors · 2021-08-10
> **Response to Reviewer pCxH**
>
> Thank you for your positive comments.

---

### Official Review · Reviewer_q6Z4 · 2021-07-14

**Rating:** 7
**Confidence:** 4

**Summary:**

This papers introduces two notions, proxy convexity and proxy PL condition, in order to prove convergence results for non-convex functions, in particular for the analysis of neural network training.
1. A function $f$ is $(g,h)$-proxy convex if $\langle \nabla f(w),w-v\rangle \geq g(w)-h(v)$ for all $w,v$. Under this condition, it's proved that the function value of $g$ will be decreased to the smallest value of $h$ after polynomial number of SGD updates.
2. A function $f$ satisfies $(g,\xi,\alpha,\mu)$-proxy PL condition if $\lVert \nabla f(w)\rVert^\alpha \geq \frac{1}{2}\mu(g(w)-\xi).$ Under this condition, it's proved that the function value of $g$ will be decreased to $\xi.$

Then the authors showed that many existing analyses of neural network training can be proved through proxy convexity or proxy PL condition.


**Limitations And Societal Impact:**

The authors have adequately addressed the limitations and potential negative societal impact of their work

**Main Review:**

This paper proposed two notions, proxy convexity and proxy PL condition, for non-convex optimization. Under these conditions, the authors proved certain convergence results on the proxy functions, which more or less follows from a standard analysis. It's interesting to see that this framework can indeed capture some seemingly pretty different analysis (deep net in the NTK regime and one-hidden-layer outside the NTK regime). So I think this framework provides a new perspective to understand the current analyses for neural network training and might be able to inspire new analysis in different settings.

That being said, my main concern for this paper is that no new neural net training results were proved under the prosed framework. I think in order to fully justify this framework, besides incorporating existing analyses to this framework, it would be more convincing if this framework allows new results to be proved in other settings. In some sense, this would mean this framework not only 'fits' the existing analyses, but also 'generalizes' to new settings.

**Time Spent Reviewing:**

4

---

> ### Author Response · Authors · 2021-08-10
> **Response to Reviewer q6Z4**
>
> Thank you for your comments.
>
> Re: new neural network training results, we wish to emphasize that in this work our aim is to put forward a new non-convex optimization framework that is well-suited to the particularities of neural network optimization, and to demonstrate that a number of recent seemingly disparate works in the area can in fact be unified through our framework.  In order to demonstrate that this framework is 'unifying', we believe it is necessary to ground our framework primarily in the results of prior work, rather than new optimization analyses.  Developing new analyses is an important future research direction, but we believe such analyses are not necessary for achieving the outcome we are aiming for with this paper.
>
> Even so, as we mention in our response to Reviewer qxiM, we provide one novel optimization result in our submission.  Example 3.4 provides an alternative proof of a result appearing in [16, Frei et al 2021].  In that work (see proof of Thm. 2.6), the authors relied upon a Perceptron-style proof technique that considers the correlation $\langle W^{(t)}, V\rangle$ between the weights of SGD and those of a fixed weight matrix $V$.  This proof technique relies heavily upon the homogeneity of the leaky ReLU activation (i.e., that $z\sigma'(z)=\sigma(z)$) and would not hold for other activation functions.  By contrast, in Example 3.4 we show that the proxy PL inequality framework allows for a guarantee for SGD for networks with smoothed leaky ReLU activations via a more typical smoothness-based optimization analysis.  Note that the Perceptron-style proof of [16] is completely different from the smoothness-based proof we used for the proof of Theorem 3.1 which is the basis of Example 3.4.

---

> > ### Comment · Reviewer_q6Z4 · 2021-08-28
> > **Thanks for the response**
> >
> > Thanks for addressing my concerns. I have raised my score.

---

### Official Review · Reviewer_qxiM · 2021-07-16

**Rating:** 6
**Confidence:** 3

**Summary:**

The Polyak-Lojasiewicz (PL) inequality is a condition on a function that the gradient is lower bounded by the in terms of the gap between the functions value and the optimal value.  This paper presents a generalized conditioned with additional parameters, including a proxy function $g$ and a notion of proxy convexity using a similar proxy function $g$. Unlike the PL inequality, the new condition can be satisfied by functions having non-optimal local minima.  A theorem bounds the the error of gradient descent (with respect to the proxy convexity function $g$).  They give some examples where guarantees with respect to the desired function can be obtained from the guarantees on the proxy function $g$.



**Limitations And Societal Impact:**

The main limitation is the obvious one of finding appropriate $g$ and $\xi$ for the generalized PL inequality.  I don't think it needs to be addressed further.



**Main Review:**

Neither the proxy convexity generalization nor the generalization of the PL inequality condition are surprising. The paper’s contribution is fitting the two together to obtain bounds on how GD approximately optimizes the proxy function $g$ (and examples where this implies good bounds with respect to the original function $f$).
The authors show how previous results git into their generalize PL framework.

The formulation appears to be useful only when stationary points are “good”.  The bounds seem less less meaningful when there are bad unstable stationary points as in the ReLU example around line 73.  Therefore the approach may be good progress towards, but not a revolutionary step, in our understanding of neural network success for non-convex functions.

Although the paper shows that many previous results fit into the proxy convexity framework, it was unclear to me if any of the examples represent new or better bounds than previously known.



**Time Spent Reviewing:**

~4

---

> ### Author Response · Authors · 2021-08-10
> **Response to Reviewer qxiM**
>
> Thank you for your comments.
>
> We agree with the reviewer that the proxy PL inequality can be inappropriate in settings where there exist stationary points with bad performance, such as the single ReLU neuron example. However, we wish to clarify that proxy convexity (and not the proxy PL inequality) can be appropriate in settings where some stationary points have bad generalization performance.   Indeed, we show in Example 4.2 that the single ReLU neuron satisfies proxy convexity, and using this we derive a strong generalization guarantee despite the existence of bad stationary points.  We believe the combined framework of proxy convexity and proxy PL inequality is the most cohesive framework for analyzing neural network optimization to date.
>
>
> Regarding the introduction of new results, there is one novel result provided in the paper which we did not emphasize.  In particular, Example 3.4 provides an alternative proof for Theorem 2.6 of [16, Frei et al 2021].  In that work (see proof of Thm. 2.6), the authors relied upon a Perceptron-style proof technique that considers the correlation $\langle W^{(t)}, V\rangle$ between the weights of SGD and those of a fixed weight matrix $V$.  This proof technique relies heavily upon the homogeneity of the leaky ReLU activation (i.e., that $z\sigma'(z)=\sigma(z)$) and would not hold for other activation functions.  By contrast, in Example 3.4 we show that the proxy PL inequality framework, via Theorem 3.1, allows for a guarantee for SGD for networks with smoothed leaky ReLU activations via a more typical smoothness-based optimization analysis.  We emphasize that the Perceptron-style proof of [16] is completely different than the smoothness-based proof we used for the proof of Theorem 3.1.  We will be sure to clearly explain this in the final version.

---

### Decision · Program_Chairs · 2021-09-27

**Decision:**

Accept (Poster)

**Comment:**

This paper introduces the notions of proxy convexity and proxy PL condition, to analyze convergence in optimization of non-convex functions. This is of particular interest in non-convex neural network training, for which a unified analysis is not yet available in the existing literature. The paper shows that these notions enable a unified analysis of convergence in various settings including the Neural Tangent Kernel (NTK), i.e., infinite-width regime, the fixed width regime in two-layer networks and learning a single ReLU neuron.

Overall, the paper is clearly written and provides good progress towards this direction, although most of the presented convergence results can be established via other methods. The reviewers all agree that the paper presents a nice framework, and expressed minor concerns and suggestions. Please take into account the updated reviews when preparing the final version to accommodate the requested changes. Thank you for your submission to NeurIPS.